# Dynamic gamma modulation of hippocampal place cells predominates development of theta sequences

**Ning Wang[1], Yimeng Wang[1], Mingkun Guo[1], Ling Wang[1,2,3], Xueling Wang[1], Nan Zhu[1], Jiajia Yang[1,2,3], Lei Wang[4]\*, Chenguang Zheng[1,2,3]\*, Dong Ming[1,2,3]\***

[1]Academy of Medical Engineering and Translational Medicine, Medical College, Tianjin University, Tianjin, China; [2]Tianjin Key Laboratory of Brain Science and Neuroengineering, Tianjin, China; [3]Haihe Laboratory of Brain-Computer Interaction and Human-Machine Integration, Tianjin, China; [4]School of Statistics and Data Science, Nankai University, Tianjin, China

## eLife Assessment

Using electrophysiological recordings in freely moving rats, this **valuable** study investigates the role of gamma oscillations in the development of spatial representations in the hippocampus. Specifically, **solid** evidence supports the claim that distinct gamma oscillatory inputs contribute to the emergence of 'theta sequences', which encode the animal's ongoing trajectory. This study will be of interest to neuroscientists working in the fields of spatial navigation and neuronal dynamics.

**Abstract** The experience-dependent spatial cognitive process requires sequential organization of hippocampal neural activities by theta rhythm, which develops to represent highly compressed information for rapid learning. However, how the theta sequences were developed in a finer timescale within theta cycles remains unclear. In this study, we found in rats that sweep-ahead structure of theta sequences developing with exploration was predominantly dependent on a relatively large proportion of FG-cells, that is a subset of place cells dominantly phase-locked to fast gamma rhythms. These ensembles integrated compressed spatial information by cells consistently firing at precessing slow gamma phases within the theta cycle. Accordingly, the sweep-ahead structure of FG-cell sequences was positively correlated with the intensity of slow gamma phase precession, in particular during early development of theta sequences. These findings highlight the dynamic network modulation by fast and slow gamma in the development of theta sequences which may further facilitate memory encoding and retrieval.

## Introduction

The hippocampus is known as its central role in encoding and storing experience and thus is essential for forming episodic memories (*Buzsáki et al., 2022*; *Cohen and Eichenbaum, 1993*). In the context of active navigation in an environment, hippocampal place cells are organized to fire sequentially within individual cycles of theta oscillations (4–12 Hz) and form time-compressed representations of behavioral experiences, referred to as theta sequences (*Colgin, 2020*; *Comrie et al., 2022*; *Dragoi and Buzsáki, 2006*; *Skaggs et al., 1996*; *Wallis, 2018*). By sweeping ahead from the past to the upcoming locations, theta sequences represent information about potential future paths that could guide goal-directed behaviors (*Gupta et al., 2012*; *Joshi et al., 2023*; *Zheng et al., 2021*). These predictive theta sequences exhibited experience-dependent development when rats were exploring

**\*For correspondence:**
lwangstat@nankai.edu.cn (LW);
cgzheng@tju.edu.cn (CZ);
richardming@tju.edu.cn (DM)

**Competing interest:** The authors declare that no competing interests exist.

a novel environment or learning a novel goal location (*Feng et al., 2015*; *Gobbo et al., 2022*; *Wikenheiser and Redish, 2015*; *Zheng et al., 2021*). However, how the hippocampal network modulated the development of theta sequences' temporospatial-compressed organization remains incompletely understood.

A common assumption was established by previous studies that theta sequences were manifestations of theta phase precession, in which an individual place cell fired at progressively earlier theta phase as the animal traversed the place field (*Aoki et al., 2023*; *Foster and Wilson, 2007*; *O'Keefe and Recce, 1993*; *Qasim et al., 2021*; *Skaggs et al., 1996*). However, when examined the theta sequences with experience at single trial level, their sequential structures were not developed on the first experience of a novel environment even if the phase precession of individual place cells has been observed (*Colgin, 2020*; *Feng et al., 2015*). This suggests that the dynamic process of theta sequences development could be modulated by another network-level mechanism with neuronal firing organized at a finer timescale within theta cycles. Gamma (25–100 Hz) could be a candidate that is thought to interact with theta to temporally organize theta sequences (*Dragoi and Buzsáki, 2006*; *Fernandez-Ruiz et al., 2023*; *Lisman and Jensen, 2013*). As two subtypes of gamma, fast and slow gamma rhythms (*Colgin et al., 2009*; *Schomburg et al., 2014*; *Zhang et al., 2019*) occur at distinct theta phases and coordinate theta sequences in different manners (*Zheng et al., 2016*). This indicates that fast and slow gamma may act as different tuners but collaborate together in modulating theta sequence development.

Fast gamma (~65–100 Hz), associated with the input from the medial entorhinal cortex, is thought to rapidly encode ongoing novel information in the context (*Fernández-Ruiz et al., 2021*; *Kemere et al., 2013*; *Zheng et al., 2016*). During fast gamma rhythms, place cell spikes occurred across all positions within a place field and displayed theta phase precession (*Bieri et al., 2014*). This type of cells was identified recently as 'phase-precessing' cells, in which the phase-precession pattern emerges with strong fast gamma (*Guardamagna et al., 2023*). Their discrete distribution of theta phases is likely to contribute to the sequential structure formation when the sequences have been developed (*Wang et al., 2020*). However, a question is raised about how these fast gamma-modulated cells be coordinated for the developmental process of theta sequences.

Slow gamma (~25–45 Hz) in the hippocampal CA1 reflects information received from CA3, which is responsible for the integration processing of learned information and experience (*Colgin et al., 2009*; *Zhu et al., 2023*). In contrast to fast gamma, spiking during slow gamma exhibited dominant theta phase-locking and attenuated theta phase precession (*Guardamagna et al., 2023*; *Zhang et al., 2019*). This concentrated distribution of theta phase could be interpreted as they are coordinated within slow gamma cycles, as a finer timescale than a theta cycle. Place cells spiking exhibited slow gamma phase precession, to generate mini-sequences in an individual theta sequence, for highly time-compressed representation of ongoing and future information (*Zheng et al., 2016*; *Zheng et al., 2021*). Thus, how fast and slow gamma rhythms coordinate place cells spiking to modulate the dynamic process of theta sequence development remains unclear.

Here, we proposed a scheme that fast gamma rhythms coordinate a subgroup of place cells for rapidly encoding and processing sensory information since from the early phase of theta sequence development (*Figure 1*). The developing process would be then dependent on the slow gamma modulation, that the spikes are fired at slow gamma phase-precessed pattern for spatial information compression within theta cycles. To test this hypothesis, we investigated how fast and slow gamma separately modulated place cells activities during the whole process of theta sequence development. We found that the fast gamma phase-locked place cells (FG-cells) were crucial for the development of theta sequences, without which the temporospatially compressed structure of theta sequences could not be developed. Furthermore, these cells exhibited slow gamma phase precession within theta cycles associated with the sweep-ahead structure of theta sequences, which occurred earlier than those cells not phase-locked to fast gamma (NFG-cells). Our findings highlight the dynamic network modulation by neural oscillations fast and slow gamma, at finer timescales within theta, in development of theta sequences which may further facilitate memory encoding and retrieval.

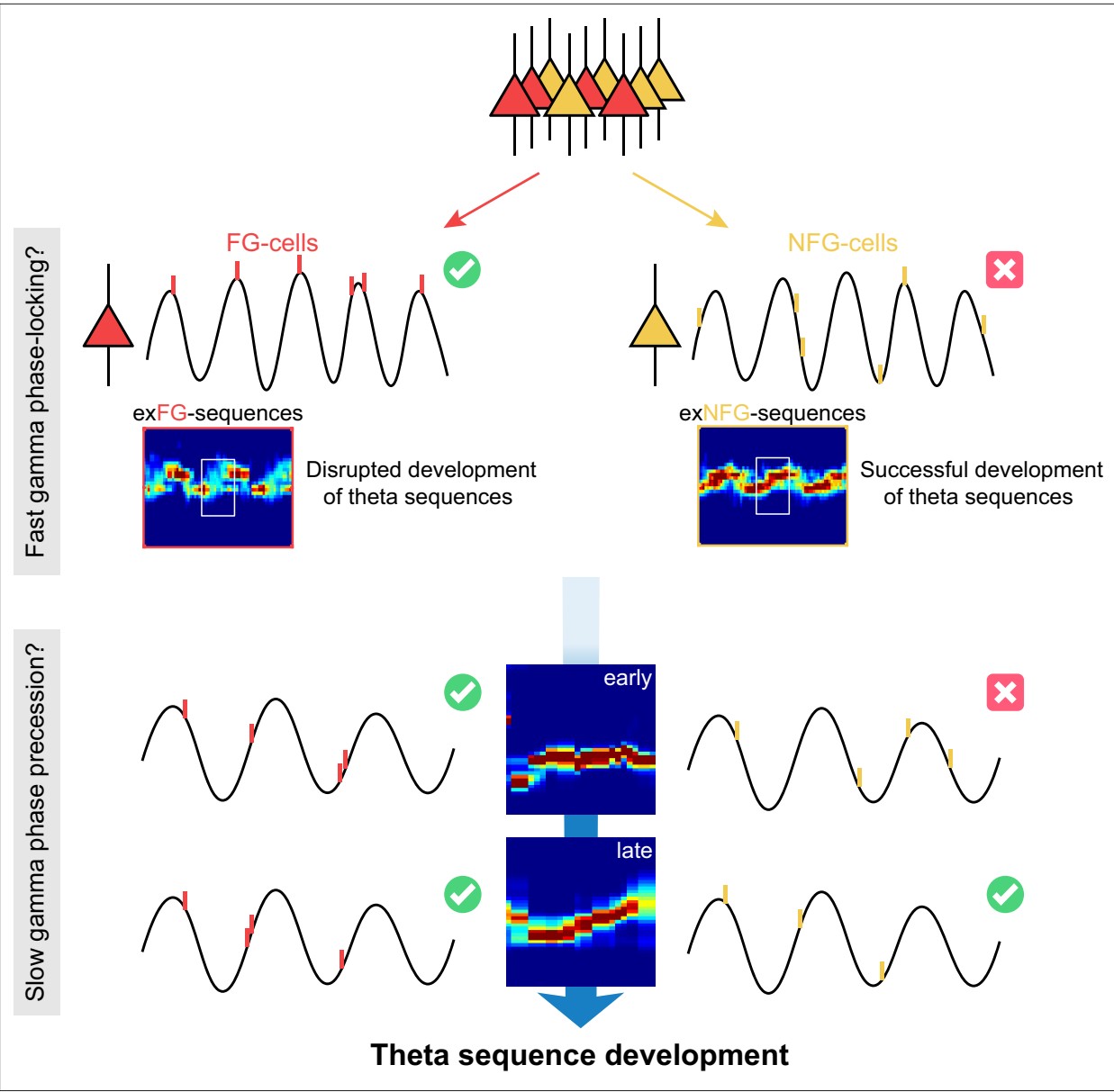

**Figure 1.** The schematic of a model for theta sequence development. Fast gamma rhythms coordinate a subgroup of place cells, as their spikes are dominantly phase-locked to fast gamma rhythms (FG-cells). The development process of theta sequences is disrupted by excluding these FG-cells. A possible model would be that development of sweep-ahead structure of theta sequences is dependent on the slow gamma modulation, that is, slow gamma phase precession, whereby spatial information could be highly compressed within theta cycles. FG-cells are those cells exhibiting slow gamma phase precession within theta cycles since from the early stage of sequence development. However, the NFG-cells only excited slow gamma phase-locking during late stage, which may unlikely contribute to the sequence development.

## Results

### A subgroup of hippocampal place cells phase-locked to fast gamma rhythms during active behaviors

To investigate the temporal organization of place cells' firing associated with the gamma rhythms during the development of theta sequences, we trained four Long-Evans rats traversed unidirectionally on a circular track, with five trials (laps) in each recording session (day). Meanwhile, neuronal spiking of place cells and local field potentials (LFPs) were recorded simultaneously in hippocampal CA1 (*Figure 2—figure supplement 1*) during active running behaviors (*Figure 2*).

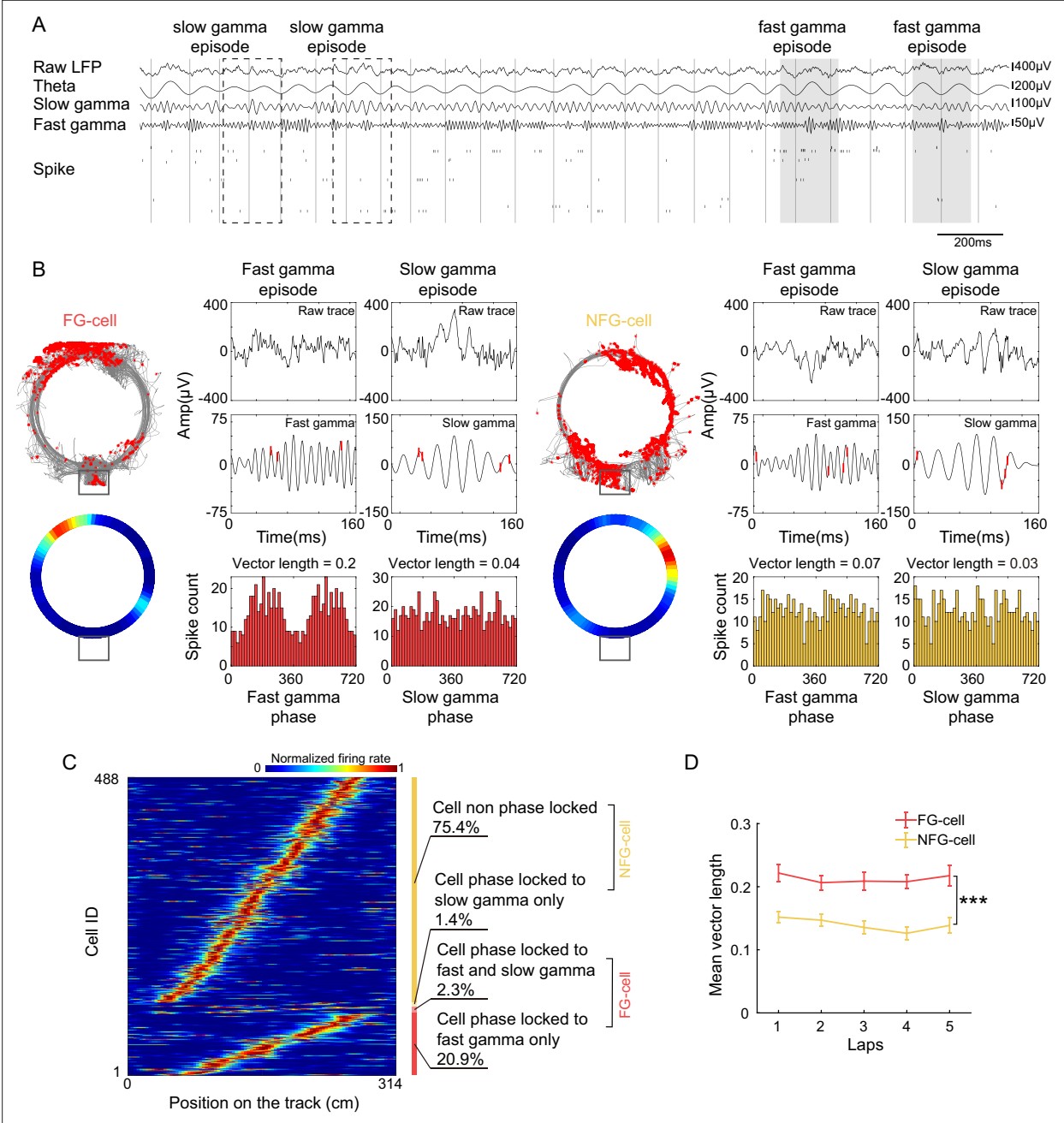

**Figure 2.** A subset of hippocampal place cells was modulated by fast gamma rhythms during active exploration. (**A**) An example of simultaneously recorded local field potentials (LFPs) (raw LFP; theta: 4–12 Hz; slow gamma: 25–45 Hz; fast gamma: 65–100 Hz) and spikes of place cells. The troughs of theta were represented by dark gray vertical lines. Slow and fast gamma episodes were marked by dashed boxes and gray blocks. (**B**) Firing characteristics of two examples of FG- and NFG-cells, including turning curve, fast and slow gamma phase distribution during detected episodes. Spikes were represented by red rasters on top of the gamma traces. Mean vector length of each cell was shown above the gamma phase distribution. (**C**) The normalized turning curves of place cells sorted by the center of mass of their main place field. The proportion of four types of cells identified according to different modulation by fast and slow gamma was shown on the right. (**D**) Mean vector length of fast gamma phase distribution of FG- and NFG-cells across successive running laps (*n* = 12 sessions). Data are presented as mean ± SEM. ***p < 0.001.

The online version of this article includes the following figure supplement(s) for figure 2:

**Figure supplement 1.** Histological verification of tetrodes.

**Figure supplement 2.** Phase-locking as a function of frequency.

**Figure supplement 3.** Theta phase precession of FG- and NFG-cells.

The slow and gamma episodes were detected in time periods when the power of each gamma type (*Figure 2A, B*, fast gamma: 65–100 Hz, slow gamma: 25–45 Hz) was above a preset threshold (see Materials and methods). We could observe both slow gamma and fast episodes occurred during active exploration. As shown in a previous study, 32% of hippocampal pyramidal cells were phase-locked to gamma rhythms during waking theta states (*Senior et al., 2008*). We further tested how the hippocampal place cells were modulated by slow and fast gamma rhythms, respectively, by quantifying the slow or fast gamma phase distributions of all spikes across all trials (*Figure 2B*). In this study, 488 place cells were included with place field distributed on the circular track but not located at the resting box. We employed local LFPs as reference to detect gamma phase-locking and found that 120 (24.6%) out of these cells were significantly phase-locked to gamma rhythms, a majority of which (113 cells, 23.2%) exhibited significant phase-locking to fast gamma rhythms (*Figure 2C*, *Figure 2— figure supplement 2A, B*). We also detected FG-cells by using LFP from a different tetrode, that is the central one of the bundle that located in the pyramidal layer, and found approximate proportion of FG-cells which phase-locked to ~75 Hz (*Figure 2—figure supplement 2C–F*, Chi-squared test, $\chi^2$ = 0.9, p = 0.4, Cramer $V$ = 0.03). Their preferred fast gamma phases (202.38 ± 2.01°) were concentrated around the peak of fast gamma cycle, which was consistent with the findings in *Schomburg et al., 2014*. However, a very small number (3.7%) of cells were found to phase-locked to slow gamma rhythms. Thus, these place cells were subdivided according to whether they were significantly phase-locked to fast gamma (FG-cells) or not (NFG-cells) during active behaviors in the subsequent analyses. Furthermore, the fast gamma phase-locking of FG-cells was stable across trials, with a significantly higher mean vector length of fast gamma phase-locking than those of NFG-cells (*Figure 2D*, repeated measure ANOVA, main effect of cell type, $F(1,22)$ = 33.9, p = 7.5 × 10⁻⁶, partial $\eta^2$ = 0.6; main effect of trials, $F(4,88)$ = 1.3, p = 0.3, partial $\eta^2$ = 0.06, no cell type × trials interaction, $F(4,88)$ = 0.5, p = 0.8, partial $\eta^2$ = 0.02). Meanwhile, we also found that a larger proportion of neurons in FG-cells had significant theta phase precession than NFG-cell (*Figure 2—figure supplement 3*, Chi-squared test, ratio of theta phase precession cell among the FG- vs. NFG-cell, n = 457 cell, $\chi^2$ = 13.7, p = 2.1 × 10⁻⁴, Cramer $V$ = 0.2). These results suggested that the FG-cells were stably modulated by fast gamma rhythms during active behaviors.

## The FG-cells were crucial for theta sequence development

As theta sequences were developed dependent on experience, we next investigated whether FG- and NFG-cells played different roles in the development of theta sequences. We then used a Bayesian decoding approach (see Methods) to identify trajectories represented as rats running on a circular track, and detected theta sequences in each single lap (*Figure 3A*). Another two decoders were built by excluding either FG-cells spikes or downsampled NFG-cells spikes, respectively, in order to compare the structures of theta sequences without each type of cells. It can be observed that posterior probability distribution of theta sequences was changed by excluding spikes from FG-cells (*Figure 3B*). The weighted correlation of each sequence (*Feng et al., 2015*) was then measured to quantify the temporospatial correlation of a theta sequence structure. We found that the weighted correlation of theta sequences decoded by excluding FG-cells spikes (exFG-sequences) was much lower than that of theta sequences decoded by excluding downsampled NFG-cells spikes (exNFG-sequences) in the same theta cycles (*Figure 3C*, generalized linear mixed model, main effect of cell type, $F(2,7255)$ = 53.0, p = 0; post hoc test, raw sequences vs. exFG-sequences, p = 0, raw sequences vs. exNFG-sequences, p = 4.6 × 10⁻⁸, exFG-sequences vs. exNFG-sequences, p = 1.1 × 10⁻⁶). This suggests that the sequential structure could be dominantly interrupted by excluding FG-cells rather than NFG-cells.

Consistent with the previous studies (*Feng et al., 2015*; *Wang et al., 2020*), we also observed the development of theta sequences along with exploration experience (*Figure 3D, E*, repeated measure ANOVA, main effect of trials, $F(4,44)$ = 7.0, p = 2.0 × 10⁻⁴, partial $\eta^2$ = 0.4). This could not be accounted for by behavioral parameters, as the rats were running at stable running speed and head direction across trials (*Figure 3F*, repeated measure ANOVA, main effect of trials: $F(4,44)$ = 0.9, p = 0.5, partial $\eta^2$ = 0.07; *Figure 3—figure supplement 1*, Watson–Williams multi-sample test, $F(4,55)$ = 0.2, p = 0.9, partial $\eta^2$ = 0.01). We further investigated whether the FG- or NFG-cells were required for the development of theta sequence structure. Interestingly, the temporospatial structure of the theta sequences cannot be developed with experience when the FG-cell spikes were excluded,

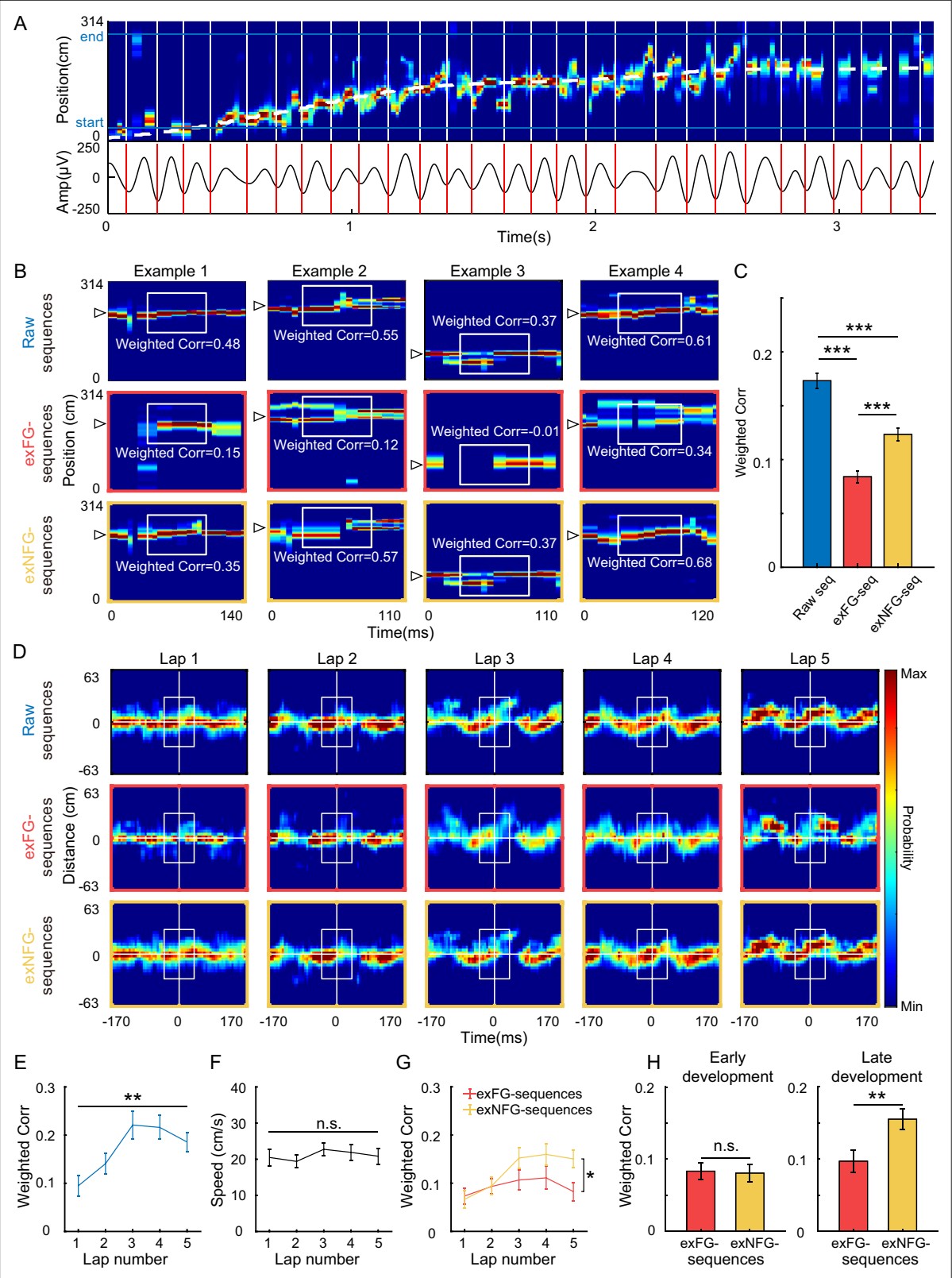

**Figure 3.** Theta sequences development was disrupted without FG-cells. (**A**) Top: An example of color-coded decoded probability with time during a single lap. The running trajectory of the animal was indicated as white dashed line. Bottom: 4–12 Hz bandpass filtered theta rhythm, with cycles divided on trough indicated as red bars. (**B**) Examples of detected theta sequences decoded by three different decoders, all spikes from all cells (top, i.e. raw sequences), all spikes excluding those from FG-cells (middle, i.e. exFG-sequences) and all spikes excluding those from NFG-cells (bottom, i.e. exNFG-

*Figure 3 continued on next page*

*Figure 3 continued*

sequences). The white triangle indicates the animal's current position. The white boxes indicate the center of sequences for calculating weighted correlation. (**C**) Weighted correlations of sequences among three types of decoders (*n* = 2423 sequences). (**D**) Averaged decoded probabilities for each lap, at a range of the animal's current position ±63 cm and the mid-time point of theta sequence ±170 ms, of a single recording session. (**E**) Weighted correlation of raw sequences was significantly increased with running laps (*n* = 12 sessions). (**F**) The running speeds were maintained across laps. (**G**) The sweep-ahead structure of the exFG-sequences was disrupted compared to that of exNFG-sequences (*n* = 12 sessions). (**H**) The effect of excluding either FG- or NFG-cells at early (left) and late (right) development of theta sequences (*n* = 24 laps). Data are presented as mean ± SEM. *p < 0.05, **p < 0.01, ***p < 0.001.

The online version of this article includes the following figure supplement(s) for figure 3:

**Figure supplement 1.** Head direction across trials.

with significantly lower weighted correlation than theta sequences constructed by excluding NFG-cell spikes (*Figure 3D, G*, repeated measure ANOVA, trials × cell type interaction, $F(4,44) = 3.0$, p = 0.03, partial $\eta^2 = 0.2$, main effect of cell type, $F(1,11) = 7.8$, p = 0.02, partial $\eta^2 = 0.4$; main effect of trials, $F(4,44) = 4.0$, p = 0.008). Since the theta sequences have been developed after Lap2, we defined the early development phase of sequences as the first two laps and late development phase of sequences as the last two laps of exploration experience, respectively. We found that the weighted correlation of both exFG- and exNFG-sequences was low during early development (*Figure 3H*, paired *t*-test, $t(24) = 0.2$, p = 0.9, Cohen's *d* = 0.04). During late development, however, the weighted correlation of exNFG-sequences was higher than that of exFG-sequences (*Figure 3H*, paired *t*-test, $t(24) = -3.7$,

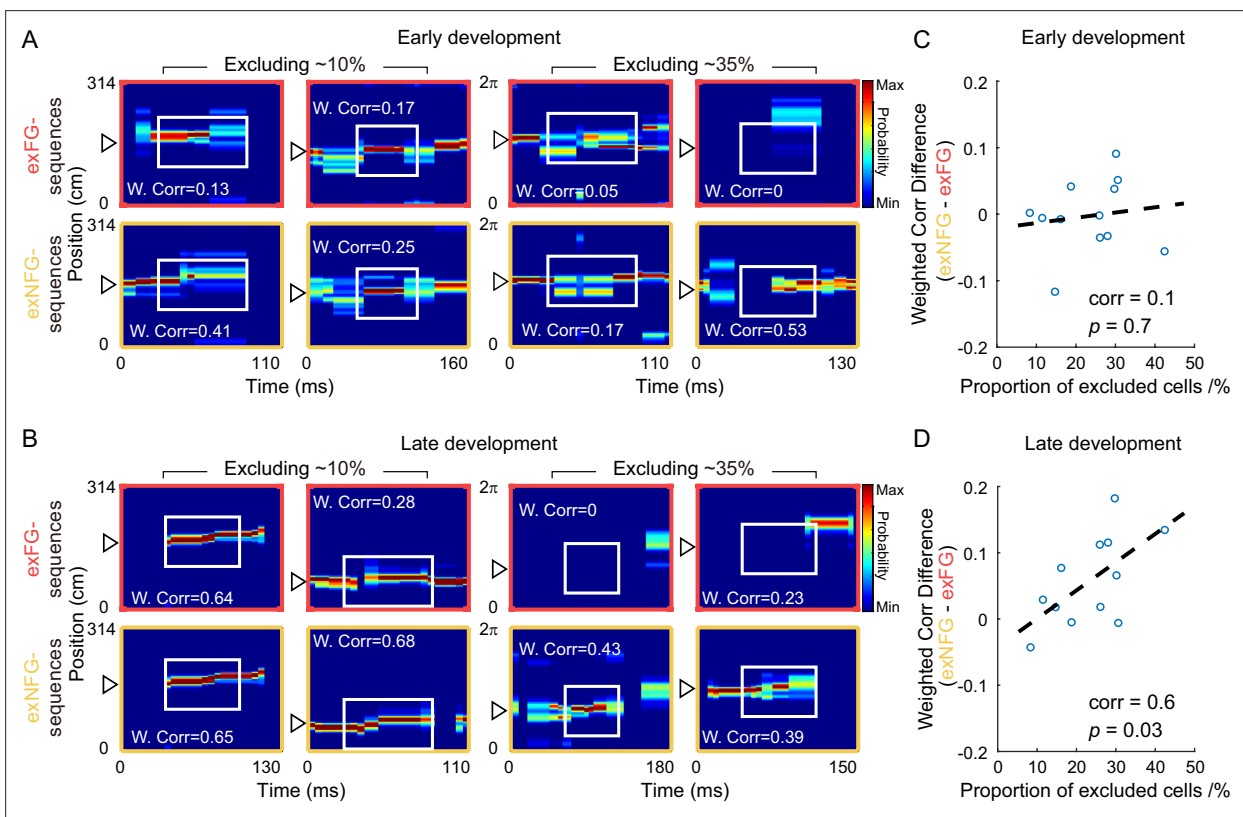

**Figure 4.** Temporospatially compressed structure of theta sequences required a relatively large proportion of FG-cells. (**A**) Examples of detected theta sequences in early development. Each column is a pair of theta sequences decoded by two different decoders, that is exFG- and exNFG-decoder, in a same theta cycle. The left two columns of sequences were detected from laps with a relatively small proportion (~10%) of FG-cells. The right two columns of sequences were detected from laps with a relatively large proportion (~35%) of FG-cells. Weighted correlation (W.Corr) of each sequence was shown. (**B**) Same as (**A**), but for late development of theta sequences. (**C**) The difference of weighted correlation between exFG- and exNFG-sequences as a function of the proportion of excluded cells, in early development. Each scatter represents data from each recording session (*n* = 12 sessions). The dashed line is the linear regression line. (**D**) Same as (**C**), but for late development of theta sequences.

p = 0.001, Cohen's *d* = 0.8). These results suggest that fast gamma modulation of place cells may contribute to the development of theta sequence with exploration experience.

In order to estimate to what extent, the FG-cells contributed to theta sequence development, we investigated the relationship between the percentage of FG-cells and the formation of sequential structure in different developments. In our dataset, the percentage of FG-cells out of all place cells varied across days, thus we detected theta sequences from trials with relatively low percentage (~10%) and high percentage (~35%) of FG-cells. The number of excluded NFG-cells for decoding was matched up with the number of excluded FG-cells by a pseudo-random method in each day. During early development, the temporospatial structure of theta sequences has not been developed by excluding either low or high percentage of each type of cells (*Figure 4A*). During late development, however, the sweeping-ahead structure of the theta sequence was significantly disrupted when excluding a high percentage (~35%) of FG-cells, which was not observed when excluding a low percentage (~10%) of FG-cells (*Figure 4B*). The sequential structure could be maintained by excluding NFG-cells at either low or high percentage. To quantify the relationship between sequence structure and the number of excluded neurons during early and late development, we then measured the weighted correlation difference between exFG- and exNFG-sequences as a function of the fraction of excluded cells. The results showed that the weighted correlation difference was close to 0 regardless

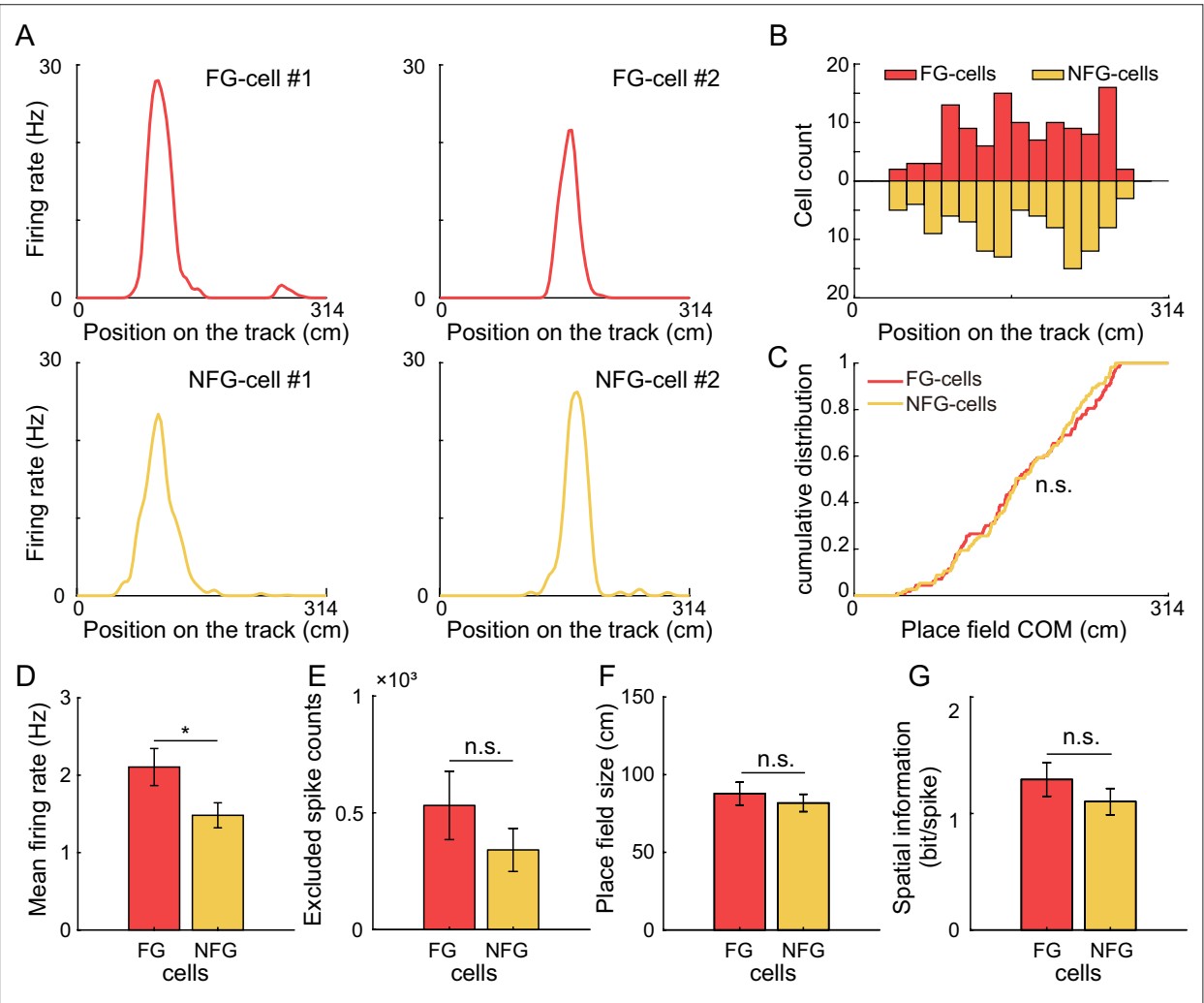

**Figure 5.** Comparison of FG- and NFG-cells firing characteristics. (**A**) Spatial tuning curves of representative FG- and NFG-cells during exploration on the circular track. Each column of cells exhibited similar place field positions on the track. (**B**) Distribution of the place field centers of mass (COM) of FG- and NFG-cells as a function of position on the track (*n* = 113 cells for FG-cells and downsampled *n* = 113 cells for NFG-cells). (**C**) Cumulative distribution of FG- and NFG-cells' place field COM. (**D**) Mean firing rate, (**E**) excluded spike counts in decoding, (**F**) place field size, and (**G**) spatial information of FG- and NFG-cells (*n* = 12 sessions). Data are presented as mean ± SEM. *p < 0.05.

of the fraction of excluded cells during early development (*Figure 4C*, Pearson's correlation, Corr = 0.1, p = 0.7). During late development, however, we found that the weighted correlation difference was positively correlated with the fraction of excluded cells (*Figure 4D*, Pearson's correlation, Corr = 0.6, p = 0.03). These results suggest different roles of FG- and NFG-cells in the development of the sweeping-ahead structure of theta sequence with experience, that the more temporospatially compressed structure of theta sequences may require a larger percentage of FG-cells.

In addition, we tested whether these results could be accounted for by the different firing characteristics between FG- and NFG-cells. We found that although these two types of cells took different proportion of all recorded place cells, the place fields of both types of cells could cover the entire track uniformly (*Figures 2C and 5A*). We measured the distribution of center of mass (COM) of main place field of FG-cells and number-matched NFG-cells (*Figure 5B*) and found similar distribution of place field COM between cell types (*Figure 5C*, two-sample Kolmogorov–Smirnov test, $Z = 0.1$, p = 0.8). Furthermore, Although the mean firing rate of FG-cell was significantly higher than NFG-cell (*Figure 5D*, Student's *t*-test, $t(22) = 2.1$, p = 0.04, Cohen's $d = 0.9$), the theta sequences in the above analyses were decoded by excluding similar numbers of spikes from FG- and NFG-cells (*Figure 5E*, excluded spike counts, $t(22) = 1.2$, p = 0.3, Cohen's $d = 0.5$), indicating that the different development patterns between exFG- and exNFG-sequences were not due to quality difference between two decoders. The two type of cells exhibited similar place fields size (*Figure 5F*), $t(22) = 0.7$, p = 0.5, Cohen's $d = 0.3$ and spatial information (*Figure 5G*) $t(22) = 1.0$, p = 0.3, Cohen's $d = 0.4$. These findings support the hypothesis that FG-cells, but not NFG-cells, may contribute to the development of theta sequences with experience.

## FG-cells exhibited slow gamma phase precession for information compression required in theta sequence development

In our previous study, we found that the place cell spikes occurred at progressively earlier slow gamma phases across slow gamma cycles in theta sequences, so-called slow gamma phase precession (*Zheng et al., 2016*). As sequences of locations were represented at different slow gamma phases, theta sequences exhibited temporospatially compressed structure representing long paths. We next investigated if the development of theta sequence structure was related to the slow gamma phase modulation of place cells. By detecting the slow gamma phases of spikes within all theta cycles, we observed that the slow gamma phases significantly shifted across successive slow gamma cycles (*Figure 6A*, multi-sample Mardia–Watson–Wheeler test, $W(4) = 369.4$, $p < 2.2 \times 10^{-16}$; post hoc test, Cycle –1 vs. Cycle 0, $W(2) = 97.5$, $p < 2.2 \times 10^{-16}$, Cycle 0 vs. Cycle 1, $W(2) = 95.4$, $p < 2.2 \times 10^{-16}$), which was consistent with the previous finding (*Zheng et al., 2016*). Then we separated out the spike activity from the early and late development, and tested if the slow gamma phase precession still existed during different developments. Interestingly, we observed slow gamma phase shifting, but not phase precession, across successive gamma cycles during early development. The distribution of slow gamma phases in Cycle 0 was similar as that in Cycle –1, then was dispersed in Cycle 1 (*Figure 6B*, multi-sample Mardia–Watson–Wheeler test, $W(4) = 18.5$, $p = 9.8 \times 10^{-4}$; post hoc test, Cycle –1 vs. Cycle 0, $W(2) = 0.8$, p = 0.7; Cycle 0 vs. Cycle 1, $W(2) = 15.5$, $p = 4.4 \times 10^{-4}$). However, the slow gamma phase precession existed during late development (*Figure 6C*, multi-sample Mardia–Watson–Wheeler test, $W(4) = 21.2$, $p = 2.8 \times 10^{-4}$; post hoc test, Cycle –1 vs Cycle 0, $W(2) = 8.8$, p = 0.01; Cycle 0 vs. Cycle 1, $W(2) = 8.2$, p = 0.02). This finding suggests that slow gamma phase precession of place cells may occur with exploring experience.

Furthermore, we investigated how the slow gamma phase precession occurred in theta sequences developed with experience. We detected theta sequences during slow gamma episodes and measured the distribution of phase shifts across gamma cycles within single unites. In the theta sequences during slow gamma episodes, we observed the slow gamma phase precession from single unit, that the place cell spikes occurred at earlier slow gamma phases in later slow gamma cycles (*Figure 7A*). To quantify the slow gamma phase precession at single-unit level, we calculated the phase shift as the phase difference within spike pairs from adjacent active slow gamma cycles (see Methods). We found that slow gamma phase difference was significantly lower than 0 between two active slow gamma cycles (*Figure 7B*, *t*-test, $t(11) = -8.8$, $p = 2.6 \times 10^{-6}$, Cohen's $d = 5.3$), as well as among 3 or ≥4 active slow gamma cycles for strict limitation of theta sequences modulated by slow gamma (3 active cycles, $t(11) = -7.3$, $p = 1.5 \times 10^{-5}$, Cohen's $d = 4.4$; ≥4 active cycles, $t(8) = -3.6$, p = 0.007, Cohen's $d = 2.5$).

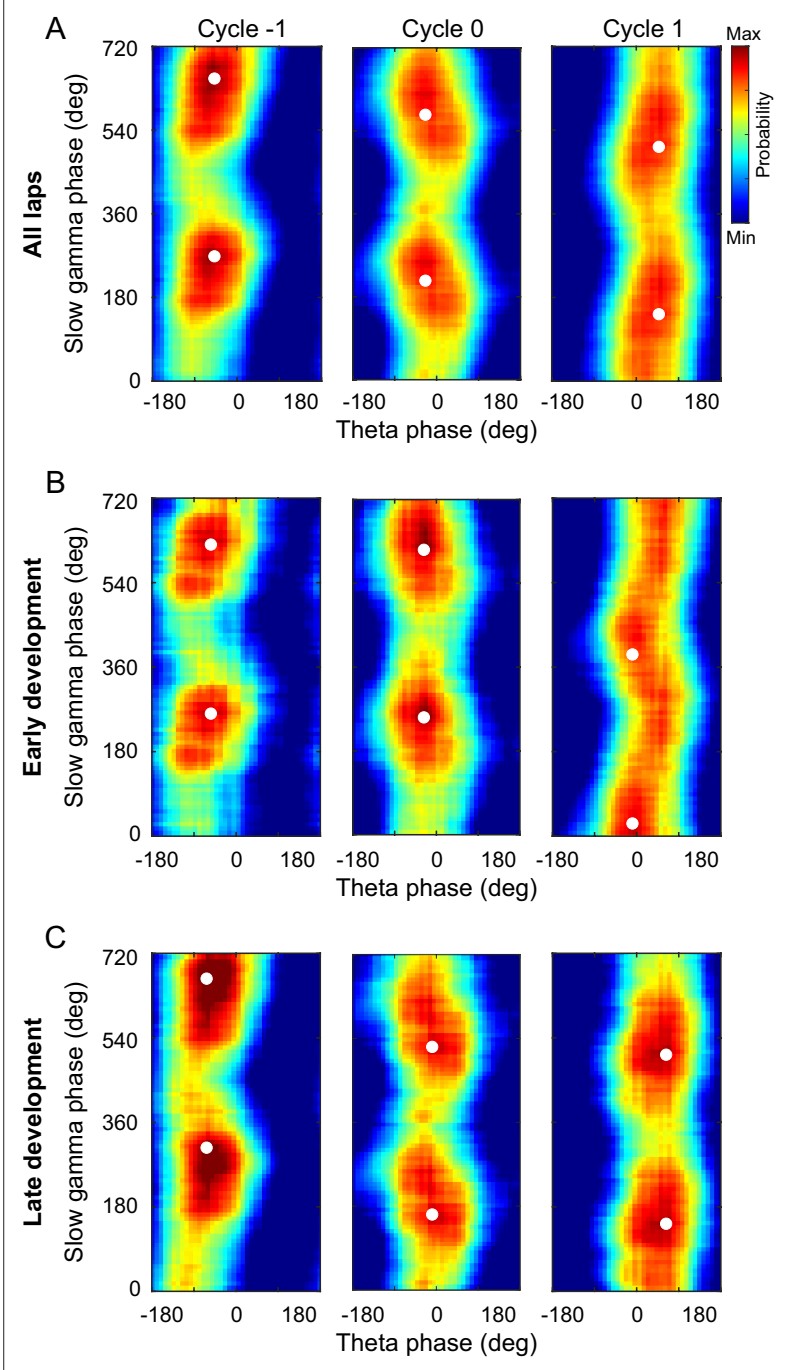

**Figure 6.** Slow gamma phase precession during early and late development of theta sequences. (**A**) Probability distributions of slow gamma phases of spikes across successive slow gamma cycles within theta sequences. Gamma cycles within theta cycles were ordered, centered at cycle 0 (i.e., gamma cycle with maximal spiking). Slow gamma phases of spikes shifted systematically across successive slow gamma cycles. (**B**) Same as (A) but during early development of theta sequences. Spikes' slow gamma phase did not significantly backward-shifted across first two slow gamma cycles. (**C**) Same as (**A**), but during late development of theta sequences. The white dots denote the peak probability of the histogram.

Furthermore, we averaged the slow gamma phase differences across different cells within a theta sequence and compare their distribution along with learning trials. In the first lap, we found that the slow gamma phase differences were not significantly different from their surrogate data (*Figure 7C*, median of phase difference from lap1 was in 95% confidence intervals of those from shuffled data, p

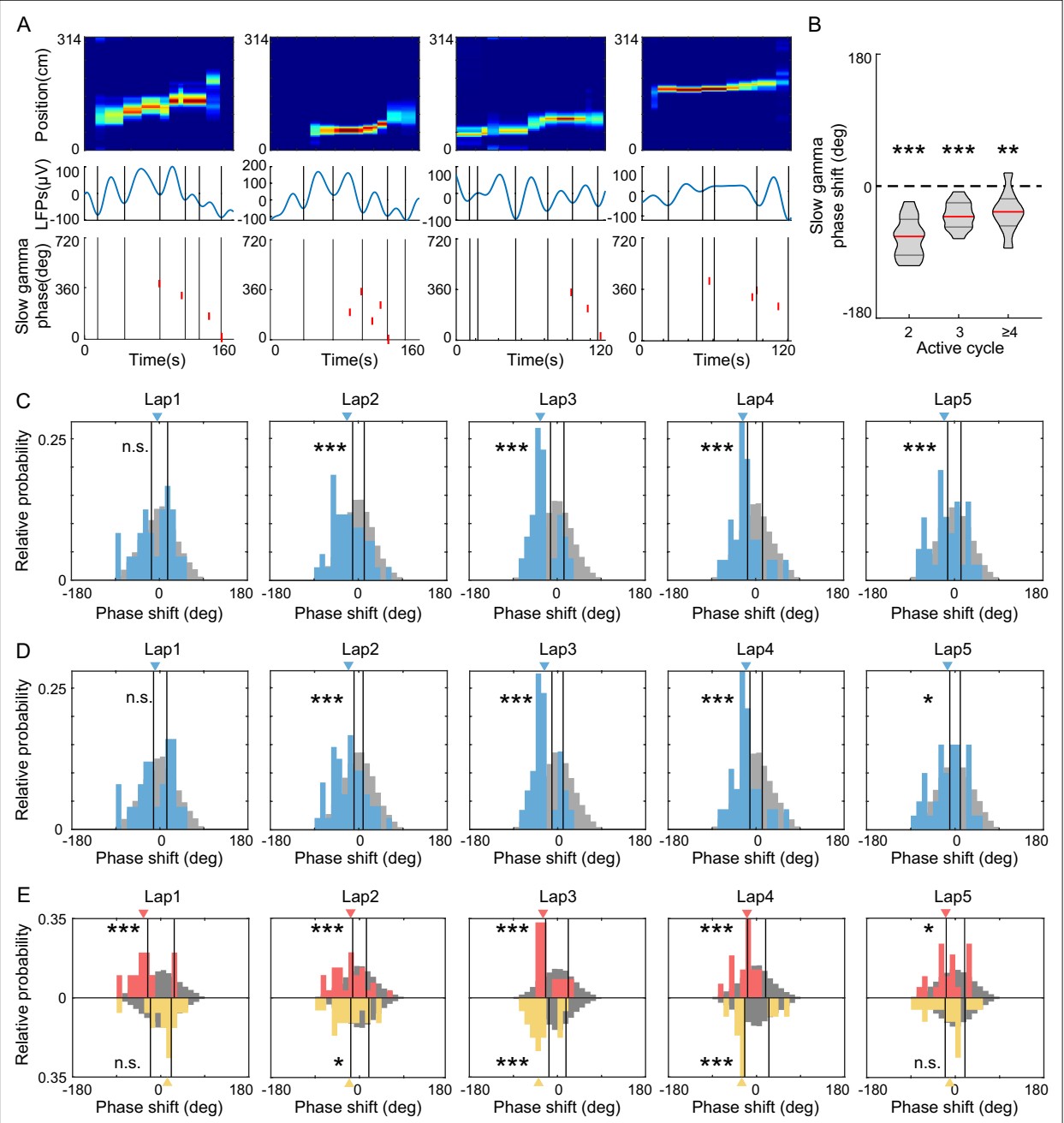

**Figure 7.** FG-cells constantly exhibited slow gamma phase precession across laps. (**A**) Examples of slow gamma phase precession within theta cycles on single-cell level. The top row shows posterior probability for theta sequences. The middle row shows the local field potentials (LFPs) waveform in the same theta cycle of the theta sequence, with each slow gamma cycle divided by black lines. The bottom row shows the slow gamma phases of spikes from a representative cell activated in this theta cycle. The spikes occurred at earlier slow gamma phase in the late slow gamma cycles. (**B**) The averaged slow gamma phase shift across *N* active cycles were significantly negative (*N* = 2, 3, or ≥4). The red lines indicate the median and the black lines indicate the 25% and 75% quantile of group data (*n* = 12 sessions for *N* = 2 and *N* = 3, *n* = 9 sessions for *N* ≥ 4). (**C**) Histogram of mean slow gamma phase shift of theta cycles across laps. The histograms of real data are shown in blue with blue triangle indicating median. The histograms of mock data are shown in gray with black lines indicating the 95th confident interval of the their median. (**D**) The same as (**C**) but for histogram of mean slow gamma phase shift of place cells across laps. (**E**) Histogram of mean slow gamma phase shift of FG-cells (red) or NFG-cells (yellow) across laps. The red and yellow triangles are the median of phase shift in FG- and NFG-cells, respectively. *p < 0.05, **p < 0.01, ***p < 0.001.

= 0.3). However, the distribution of slow gamma phase difference was negatively biased compared with their surrogate data (*Figure 7C*, median of phase difference from lap2–5 was beyond the 95% confidence intervals of those from shuffled data; lap2 p = 0.001, lap3 p = 0.001, lap4 p = 0.002, lap5 p = 0.002). These results can also be replicated by averaging the slow gamma phase differences from a single unit across successive theta sequences (*Figure 7D*, lap1 p = 0.1, lap2 p = 0.001, lap3 p = 0.001, lap4 p = 0.002, lap5 p = 0.01).

These findings raised a question of whether the development of slow gamma phase presession existed for both FG- and NFG-cells. To address this question, we quantified the distribution of slow gamma phase difference in FG- and NFG-cells, respectively. We found that the slow gamma phase difference of FG-cells consistently exhibited a stable phase precession across trials, independent with learning experience (*Figure 7E*, lap1 p = 0.007, lap2 p = 0.006, lap3 p = 0.007, lap4 p = 0.002, lap5 p = 0.02). In contrast, the slow gamma phase precession of NFG-cells was experience dependent, with negative biased distribution of phase difference occurring from lap2 (*Figure 7E*, lap1 p = 0.9, lap2 p = 0.01, lap3 p = 0.001, lap4 p = 0.009, lap5 p = 0.1). These results suggest that FG- and NFG-cells have different dynamic modulated by slow gamma rhythms during exploration experience. The slow gamma modulation of FG-cells may occur earlier than that of NFG-cells, thereby driving the location information to be coded within theta cycles in a highly temporospatially compressed way.

## FG- and NFG-cells exhibited different dynamics of slow gamma modulation along with theta sequence development

Finally, we wondered if there was correlation between slow gamma phase precession and the development of theta sequence structure, and how the FG- and NFG-cells coordinated during this process. Within each theta sequence, we measured its weighted correlation of the sequence structure, and plot it as a function of slow gamma phase difference across slow gamma cycles. Interestingly, we found most data were distributed in the Quadrant1 (Q1) compared to the shuffled data distributed at the center (*Figure 8A, B*). The theta sequences with negative slow gamma phase difference and positive weighted correlation were significantly dominant (*Figure 8C*, the probability of observed data from lap1 exceeded 95% confidence intervals of those from shuffled data, p = 0.001). This finding supports our hypothesis that the theta sequence development could be associated with the slow gamma phase precession.

We further investigated the contribution of FG- and NFG-cells to the slow gamma-modulated theta sequence development during different exploring phases. FG-cell-dominant theta sequence (FG-cell sequence) was defined as the theta sequence with at least one FG-cell fired. And NFG-cell-dominant theta sequence (NFG-cell sequence) was defined as the theta sequence with no FG-cell fired. Importantly, the distribution of FG-cell sequences was significantly biased to Q1 than that of the shuffled data, not only in late development when the theta sequences have been developed, but also in early development (*Figure 8D–F*, early development, p = 0.001; late development, p = 0.004). However, for NFG-cell sequences, their distribution was significantly biased to Q1 than that of the shuffled data only in late development (*Figure 8G–I*, bottom, p = 0.046), but not in early development (*Figure 8G–I*, top, p = 0.08). Taken together, these results support the hypothesis that FG- and NFG-cells may exhibit different dynamical characteristics in gamma modulation, thus their coordination may contribute to the development of theta sequences. FG-cells were consistently modulated by slow gamma phase precession, which would be precondition for theta sequence development.

## Discussion

In current study, we employed a circular track exploration task to investigate the different roles of fast and slow gamma rhythms in the coordination of neurons during the experience-dependent development of theta sequences. We first found that a subgroup of place cells (FG-cells) that were phase-locked to fast gamma played a crucial role in the development of theta sequences. By excluding a sufficiently high proportion of FG-cells, the sweep-ahead structure of theta sequences was significantly disturbed compared to that of excluding equivalent number of NFG-cells. These FG-cells exhibited slow gamma phase precession within single theta cycles throughout the developing process of theta sequences. Finally, we showed positive correlation between the intensity of slow gamma phase precession and development of the sweep-ahead structure of theta sequences.

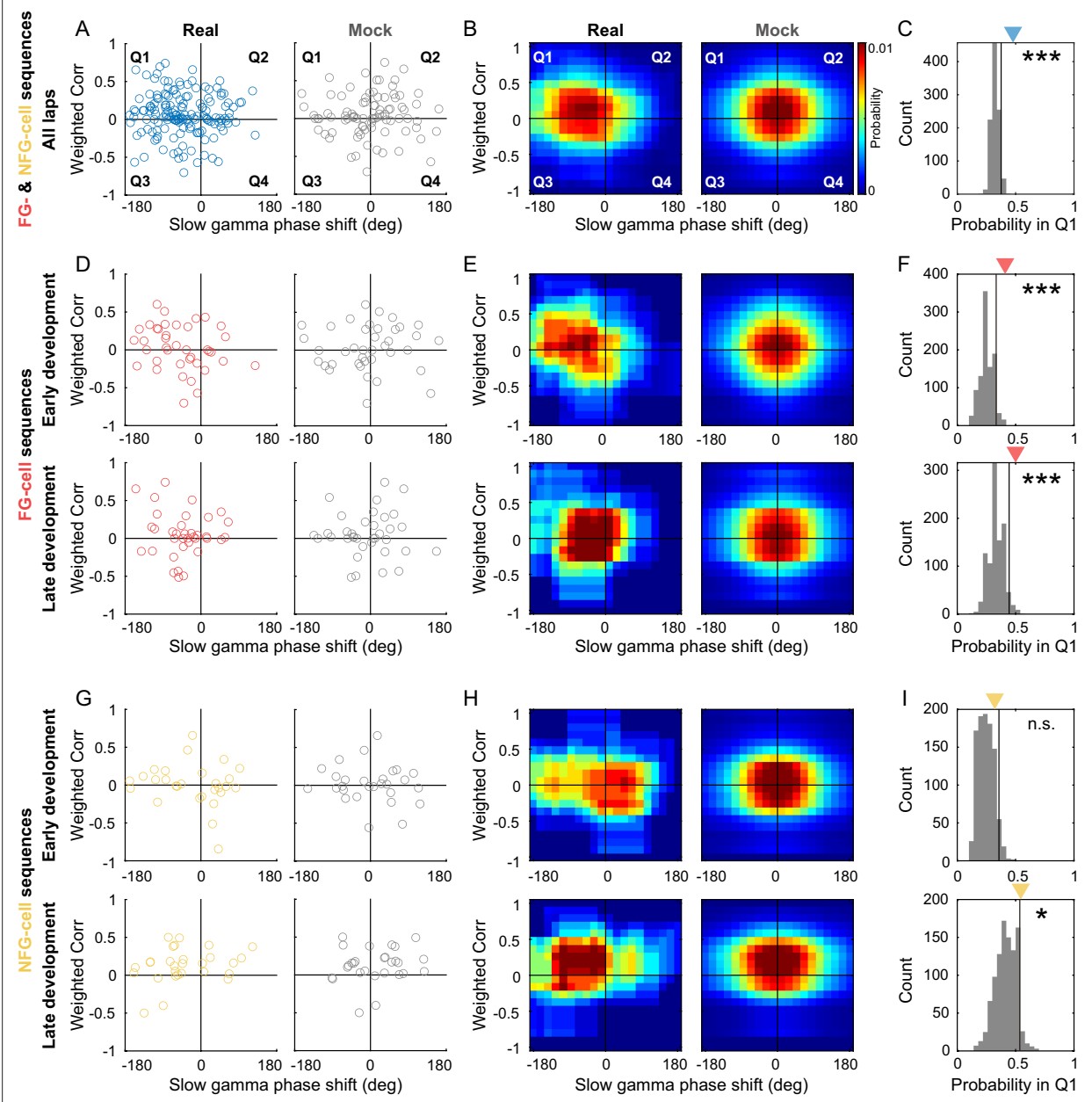

**Figure 8.** The sweep-ahead structure of FG-cell sequences was positively correlated with the intensity of slow gamma phase precession during both early and late sequence development. (**A**) Scatter plot of averaged slow gamma phase shift as a function of weighted correlation of sequences during all laps. Left panel shows real data (blue) and right panel shows mock data (gray). (**B**) Probability distributions of sequences falling in four quadrants (Q1–Q4) with slow gamma phase shift (<0 or >0) and weighted correlation (<0 or >0). (**C**) The count of sequences falling in Q1 (blue triangle) was significantly higher than 95% quantile of the shuffled data (black line). The gray histogram indicates the relative probability of mock data falling in Q1 by 1000 times shuffling. (**D**) Scatter plot of averaged slow gamma phase shift as a function of weighted correlation of FG-cell sequences during early development (top) and late development (bottom). (**E**) Probability distributions of FG-cell sequences falling in four quadrants during early development (top) and late development (bottom). (**F**) The count of FG-cell sequences falling in Q1 (red triangle) was significantly higher than 95% quantile of the shuffled data (black line) during both early development (top) and late development (bottom) of sequences. (**G, H**) Same as (**D, E**), but for NFG-cell sequences. (**I**) The count of NFG-cell sequences falling in Q1 (red triangle) was significantly higher than 95% quantile of the shuffled data (black line) only during late (bottom) but not early (top) development of sequences. *p < 0.05, ***p < 0.001.

Fast gamma episodes were detected with dominantly high probability as initial exploration of each day and decreased probability as being familiar to the environment (*Bieri et al., 2014*). This could be related to the feedforward communication from upstream to downstream regions by using fast gamma frequencies (*Vinck et al., 2023*). Although the information propagation of visual

stimulation was unlikely synchronized between visual cortex and hippocampal CA1 (*Schneider et al., 2023*), the MEC inputs could also be entrained by fast gamma to process ongoing sensory information to the hippocampus (*Buzsáki and Schomburg, 2015*; *Colgin et al., 2009*). Thus, the fast gamma phase-locked ensemble may act as a receiver of novel information that organized its intrinsic individual neurons in a time-compressed and spatially ordered sequence to integrate the information under the mesoscopic cellular substrate modulated by fast gamma rhythms (*Fernandez-Ruiz et al., 2023*). These neurons exhibited similar firing characteristics as those of fast gamma non-phase-locked neurons, however, they showed higher probability presenting theta phase precession. Therefore, this fast gamma phase-locked ensemble likely has overlapping with 'bimodal cells' involved in the formation of theta sequence structure including reverse and forward windows (*Chu et al., 2023*; *Wang et al., 2020*). In this case, regardless of these neurons, the sweep-ahead structure of theta sequences could not be developed with learning, with the extent of structure impairment positively correlated with the proportion of excluded fast gamma phase-locked neurons.

We noted that a relative low proportion of gamma phase-locked cells, particularly slow gamma phase-locked cells, was detected in this study. A potential reason would be that the local LFP was chosen as a reference for estimating the phase-locking, not a reference from specific stratum lacunosum-moleculare or stratum radialis which exhibited stronger gamma episodes (*Belluscio et al., 2012*; *Fernández-Ruiz et al., 2017*; *Schomburg et al., 2014*). This may also cause spike leakage to constraint dominant fast gamma phase-locking (*Csicsvari et al., 2003*). However, this would little affect the main conclusion because FG-cells were also detected by using LFP from a different tetrode, that is the central one of the bundle that located in the cell body layer, and found approximate proportion of FG-cells (*Figure 2—figure supplement 2*). In addition, rats were freely moving in a relatively familiar environment, and were not required to perform in a complex spatial memory task in a novel context which may lead to reduced proportion of gamma-modulated neurons (*Kitanishi et al., 2015*). Additionally, our recordings were almost exclusively conducted from the deep CA1 layer where FG-cells were obtained, consistent with the dominant ~50–100 Hz gamma power driven by entorhinal cortex layer III (EC3) inputs to stratum lacunosum-moleculare (*Belluscio et al., 2012*; *Colgin et al., 2009*; *Fernández-Ruiz et al., 2017*). The contribution of FG-cells in theta sequences likely reflects intrinsic EC3-CA1 circuit properties in spatial coding.

Theta phase precession has long been recognized as the only cause of forming the dominant theta sequence structure. The current findings offered some possible additional insights to this point. Our finding that FG-cells displayed stronger theta phase precession than NFG-cells was consistent with previous finding (*Guardamagna et al., 2023*), that the theta phase precession pattern emerged with strong fast gamma. Since slow gamma phase precession occurred within theta cycles, it is hard to consider the contribution of these factors to theta sequences development, without taking theta phase precession into account. But one should be noted that the theta sequences could not be developed even if theta phase precession existed from the very beginning of the exploration (*Feng et al., 2015*). These findings suggest that theta phase precession, together with other factors, impact theta sequence development. But the weight of these factors and their interactions during memory and learning remain important aspects of further investigation.

The important role of fast gamma in modulating theta sequence development was also supported by a specific perturbation experiment. Liu and colleagues optogenetically entrained the MEC GABAergic cells at an artificial gamma frequency (53 Hz) and observed the loss of phase precession patterns in CA1 place cells, and thereby the disrupted development of theta sequences (*Liu et al., 2023*). The failure of sequential firing organization during behaviors impaired the formation of a predictive map related to the highly time-compressed awake replay sequences executing information storage. Fast gamma was also found diminished between MEC and CA1 in transgenic model mice of Alzheimer's Disease, linked with spatial memory impairment in AD as a circuit mechanism (*Jun et al., 2020*). On the other hand, our previous study showed that fast gamma-dominated theta sequences represented shorter trajectories with relative less spatial information within a theta cycle (*Zheng et al., 2016*). This is indeed not contradictory with the current results, as we found that slow gamma, related to information compression, was also required to modulate fast gamma phase-locked cells during sequence development. We replicated the results of slow gamma phase precession at ensemble level (*Zheng et al., 2016*), and furthermore observed it at late development, but not early development,

of theta sequences. Thus, a scenario could be drawn out that the network modulator was transitioned from fast gamma to slow gamma throughout the sequence development with learning.

At single neuron level, we also observed that spikes of an individual place cell occurred at precessed slow gamma phases across consecutive slow gamma cycles under the time-scale of a theta window. At a finer scale within a theta cycle, slow gamma cycles coordinated place cells in a 'mini-sequence' way (*Zheng et al., 2016*), in order to integrate information of long spatial trajectory related to the memory retrieval and predictive behaviors. This negatively biased phase difference distribution appeared from lap2, when the sweep-ahead structures of theta sequences have not been fully developed, suggesting that slow gamma phase precession of place cells contributed to the information compression required to the theta sequence development. More interestingly, we found that slow gamma phase precession of a subset of neurons was absent at initial exploration when the theta phase precession has already displayed (*Feng et al., 2015*). This finding was in line with a previous study showing that silencing of CA3 output did not affect theta phase precession of a single cell, but impaired theta sequence structures (*Middleton and McHugh, 2016*). A possibility was that recruitment of PV-interneurons during the CA3-driven descending phase of the theta cycle, associated with the induction of slow gamma oscillations in CA1, was found reduced in the absence of CA3 input (*Fernandez-Ruiz et al., 2023*; *Topolnik and Tamboli, 2022*). We also provided additional evidence about the relationship between slow gamma phase precession and theta sequence formation, that the more negative slow gamma phase difference across cycles, the more sweep-ahead structure represented at late development of theta sequences.

Our results supported such a possible model of gamma modulation to mediate theta sequence development (*Figure 1*). The theta sequences were linked to a balance between fast and slow gamma modulations, especially in FG-cells, which displayed both fast gamma phase-locking and slow gamma phase precession. As shown in previous work (*Zheng et al., 2016*), slow gamma phase precession supported spatial information retrieval in a temporally compressed format. Thus, we expected slow gamma precession in all cells during later laps when spatial information retrieval occurs. However, during early laps, when novel information was being encoded, only FG-cells exhibited slow gamma precession. This may suggest that FG-cells help manage the balance between encoding and retrieval through different phase-coding mechanisms: fast gamma for real-time encoding and slow gamma for retrieval. This dual-modulation in FG-cells may be crucial for the development of theta sequences, allowing them to include both FG- and NFG-cells in a well-sweep-ahead pattern during later laps.

It is noteworthy that the subset of neurons (FG-cells) was modulated by both fast gamma and slow gamma rhythms during the theta sequence development (*Figure 2*), which was consistent with previous studies (*Fernández-Ruiz et al., 2017*; *Lasztóczi and Klausberger, 2016*; *Sharif et al., 2021*). These cells are seemingly challenging the above framework. However, these place cells, associated with both memory encoding and retrieval, seem to function like memory engram-like representations (*Goode et al., 2020*; *Josselyn and Tonegawa, 2020*; *Lamothe-Molina et al., 2022*; *Pettit et al., 2022*). Indeed, the proportion of c-Fos-positive place cells (putative engram cells) had higher proportion being phase-locked to fast gamma than the c-Fos-negative place cells (*Tanaka et al., 2018*). Another possibility was that these FG-cells were largely overlapped with the previously defined 'fast-firing neurons', which remained stable during learning process (*Grosmark and Buzsáki, 2016*). In contrast, the NFG-cells exhibited experience-dependent phase coding, that they likely contributed to the path representation at the late learning process. Thus, further studies would be needed to investigate how memory engram cells contribute to theta sequence development and facilitate spatial memory and navigation.

## Materials and methods
### Subjects
Four male Long-Evans rats weighing 500–700 g (~6–12 months old) were used in this study. Rats were housed in custom-made acrylic cages (40 cm × 40 cm × 40 cm) on a reverse light cycle (Light: 8:00 pm to 8:00 am). All waking behavior recordings were performed during the dark phase of the cycle (i.e., from 8:00 am to 8:00 pm). Rats were handled prior to recording drive implantation surgery. Rats recovered from surgery for at least 1 week before behavioral training resumed. Behavioral training was initiated after at least 1 week of recovery from implantation surgery. Rats were physically deprived

during experimental data collection so that their body weight was maintained at ~90% of their free-feeding body weight. All experiments were conducted according to the guidelines of the Animal Care and Use Committee of Tianjin University.

### Behavioral paradigm

Following the postsurgery recovery period, rats were trained to run on a circular track from our previous study (*Wang et al., 2024*; *Zheng et al., 2021*). In each recording day (session), rats ran five trials (or laps) on the track (inner diameter of 90 cm, height of 50 cm, and width of 10 cm, surrounded by a blue curtain) without receiving any food reward. Each trial was starting and ending at a wooden rest box attached to the track (starting point: 33 cm, ending point: 281 cm away from the box).

### Surgery and tetrode implanting

Multi-tetrode drives containing 21 independently moveable tetrodes (18 recording tetrodes and 3 reference tetrodes) were implanted into the hippocampal dorsal CA1 region (anterior–posterior [AP] –3.5 mm, medial–lateral [ML] 3.0 mm, dorsal–ventral [DV] 1 mm on day of implantation). Bone screws were placed in the skull, and the screws along with the base of the drive were covered with dental cement to affix the drive to the skull. Two screws above the cerebellum in the skull were connected to the recording drive ground. Before surgery, tetrodes were built from 17 μm polyimide-coated platinum-iridium (10/90%) wire (California Fine Wire, Grover Beach, California). The tips of tetrodes designated for single-unit recording were plated with platinum to reduce single-channel impedances under 300 kOhms. Tetrodes were gradually lowered to their target locations. Recording tetrodes were targeted to the dCA1 stratum pyramidale. One reference tetrode from each region was designated as the reference for differential recording and remained in a quiet area of the cortex throughout the experiments. This reference tetrode was moved up and down until a quiet location was found and was continuously recorded against ground to ensure that it remained quiet throughout data collection. The other reference tetrode in the dHPC was placed in the dCA1 apical dendritic layer to monitor and record LFPs in the hippocampus during placement of the other tetrodes and to later obtain simultaneous recordings from a dendritic layer.

### Data acquisition

Data were acquired using a Digital Lynx system and Cheetah recording software (Neuralynx, Bozeman, Montana). LFPs from one channel within each tetrode were continuously recorded in the 0.1–500 Hz band at a 2,000 Hz sampling rate. Input amplitude ranges were adjusted before each recording session to optimize resolution and prevent signal saturation. Input ranges for LFPs were typically within the range of ±2,000 μV to 3000μV. To detect unit activity, signals from each channel in a tetrode were bandpass filtered from 600 to 6000 Hz. The spike was detected when the signal on one or more of the channels exceeded a threshold set at 55 μV. Detected events were acquired with a 32,000 Hz sampling rate. Signals were recorded differentially against a dedicated reference channel (see 'Surgery and recording drive implantation' section). The video was recorded through the Neuralynx system with a resolution of 720 × 480 pixels and a frame rate of 25 frames per second. The animal position was tracked via a red light-emittingdiode (LED) on one of the three HS-27-Mini-LED headstages (Neuralynx, Bozeman, Montana) attached to the hyperdrive.

### Estimation of running speed and head direction

The running speed ($v_t$) at a time point ($t$) was estimated by calculating the distance between the preceding position ($x_{t-1}$, $y_{t-1}$) and the following position ($x_{t+1}$, $y_{t+1}$), and dividing by the elapsed time (2 × 1/position sampling frequency). The sampling frequency of the position data was 25 Hz, yielding a temporal resolution of 2/25 s (*Figure 3F*).

The head direction ($d_t$) at a given time point ($t$) was calculated as the arctangent of the vector formed by the preceding ($x_{t-1}$, $y_{t-1}$) and the following ($x_{t+1}$, $y_{t+1}$) position:

$$d_t = \tan^{-1} \frac{\left(x_{t+1} - x_{t-1}\right)}{\left(y_{t+1} - y_{t-1}\right)}$$

Angles were adjusted to the range of 0–360° (*Figure 3—figure supplement 1*). If the animal did not move at the time point ($t$), $d_t$ was set to $d_{t-1}$.

## Spike sorting and turning curve analysis

Detected spikes that occurred during trials were manually sorted using graphical cluster-cutting software (MClust; A.D. Redish, University of Minnesota, Minneapolis). Clusters of spikes were sorted using two-dimensional projections of three different features of spike waveforms (i.e., energies, peaks, and peak-to-valley differences) from four channels. Individual cells were identified as putative excitatory neurons that they had pyramidal cell-like waveform shape, at least 1-ms refractory period, and their clusters were perfectly separated with other clusters or noise in at least three projections. Units with mean firing rates that were higher than 15 Hz were identified as putative fast-spiking interneurons and were excluded.

The turning curve of each unit was built at 90 angle bins (approximately 3.5 cm per bin) on the circular track by using spikes during locomotion (running speed >5 cm/s, **Figure 5**). The field size of the main place field was defined as the size of contiguous position bins with a firing rate >10% of the peak firing rate. Only units with peak firing rates over 1 Hz and main field size over 3 angle bins (approximately 10.5 cm) during trials were defined as putative neurons with spatial representation. A putative excitatory neuron, which at least had one place field, was defined as a place cell. Finally, we accepted 488 place cells for further analysis.

## Spatial information score

Spatial information was calculated as previously described (**Skaggs et al., 1992**). Briefly, for each cell, the spatial information score in bits per spike was calculated from the recording during test trials (**Figure 5**), as:

$$\text{Spatial information} = \sum_i P_i \frac{\lambda_i}{\lambda} \log_2 \frac{\lambda_i}{\lambda}$$

where $\lambda_i$ is the mean firing rate of a unit in the $i$th bin, $\lambda$ is the overall mean firing rate, and $p_i$ is the probability of the animal being in the $i$th bin (occupancy in the $i$th bin/total occupancies bin).

## Detection of gamma episodes

LFP signals from each tetrode were band-pass filtered for slow gamma (25–45 Hz) and fast gamma (65–100 Hz) rhythms. Time-varying power for slow gamma and fast gamma was computed, using a Morlet's wavelet transform method described previously (**Tallon-Baudry et al., 1997**; **Zheng et al., 2016**), and then averaged within each frequency range. To detect slow and fast gamma episodes, we $z$-scored the time-varying power for slow and fast gamma across time within each trial. The time-stamp of $z$-scored time-varying power ≥3 was set as the center of a slow or fast gamma episode, and a 160-ms window around this center was defined as a gamma episode. For the windows with overlapping time bins, the one with the lower gamma amplitude was discarded. Also, the slow (or fast) gamma episode with $z$-scored fast (or slow) gamma power ≥3 was removed, ensuring that fast and slow gamma episodes are mutually exclusive (**Figure 2A, B**).

## Detection of FG- and NFG-cells

The spikes of each place cell recorded during freely running on the track were included. The time-varying slow or fast phases of local LFPs from the same tetrode with recorded cells were calculated by Hilbert transform of the bandpass filtered signals. Each spike time was matched to the closest LFP time stamp to determine its corresponding slow and fast gamma phase, and the slow and fast gamma phase distribution was established for each cell (**Figure 2B**). Mean vector length was quantified based on the slow and fast gamma phase distributions. Significant phase-locking to either slow or fast gamma was detected by Rayleigh test on phase distributions with at least 10 spikes in each lap. Because of a relatively high proportion of significant fast gamma phase-locked cells, we focused on these cells in this study and defined them as FG-cells (**Figure 2C**). Meanwhile, NFG-cells were defined as place cells which were not significantly phase-locked to fast gamma with at least 10 spikes in each lap. In addition, regarding the contribution of spike leakage to the local LFPs, we also selected an individual tetrode which located at stratum pyramidale and at the center of the drive bundle for each rat. We detected a similar proportion of FG-cells by using LFPs on this tetrode, compared with that using local LFPs (Chi-squared test, $\chi^2 = 0.9$, p = 0.4, Cramer $V = 0.03$).

### Theta phase precession

LFPs from each tetrode were filtered through a 4- to 12-Hz bandpass filter. Theta phases were estimated using angles of their Hilbert-transformed signals for each channel. Theta phase precession was examined in linearized spatial turning curves (see Spike sorting and turning curve analysis). For spikes within each place field, phase precession was computed using a circular-linear fit (*Kempter et al., 2012*). The p-value from the circular-linear regression was reported (*Figure 2—figure supplement 3*).

### Single-trial Bayesian decoding analysis

In the decoding analysis, we included the recording sessions with 44 ± 1 HPC place cells. To translate ensemble spiking activity into radian positions on the circular track, the Bayesian decoding algorithm was implemented. For each trial, a memoryless Bayesian decoder was built at 90 angle bins (ranging from 0 to 2pi) by using spikes during locomotion (running speed >5 cm/s) in this trial. Decoding was performed using a 20-ms sliding time window with 5-ms stepping. For each time window, the number of spikes occurring is denoted as $n_i$, so that the vector of all spiking activity simultaneously recorded for the cell is $\boldsymbol{n} = \{n_1, n_2, \cdots, n_i\}$. The probability of a rat being at position $x$, given the number of spikes $\boldsymbol{n}$ from each unit recorded in a time window, was estimated by Bayes' rule:

$$P\left(x|\boldsymbol{n}\right) = \frac{P\left(\boldsymbol{n}|x\right) \times P\left(x\right)}{P\left(n\right)}$$

where $P\left(\boldsymbol{n}|x\right)$ was approximated from the single directional firing turning curve of each unit. The approximation assumed that the turning curve of individual units was statistically independent and the number of spikes from each unit followed a Poisson distribution. Prior knowledge of position $P\left(x\right)$ was set to 1 to avoid decoding bias to any particular location on the track. The normalizing constant $P\left(n\right)$ was set to ensure $P\left(x|\boldsymbol{n}\right)$, or posterior probability, summed to 1 (*Figure 3*). The decoding position was is the set of x that maximizes $P\left(x|\boldsymbol{n}\right)$. For excluded cell decoding, position reconstruction was performed by removing the FG- or NFG-cells from $\boldsymbol{n}$, respectively. Bayesian decoding analyses were performed using custom routines in MATLAB 2020b.

### Theta sequences detection

Theta rhythms were detected from a central tetrode of all recording tetrodes which located in the stratum pyramidale of hippocampus. We cut sequences at each theta phases (0°, 10°, 20°, …, 350°) and quantified averaged weighted correlation corresponding to these 36 possible phases (see 'Weighted correlation of theta sequences'). Then, the optimal cutting phase of the theta cycle was determined when the weighted correlation of theta sequences was maximal, whereby the sequences were separated within each theta cycle (*Figure 3A*).

Raw theta sequences were detected by the following criteria: (1) during the sequence, the animal was running between the starting point and ending point of the track, with running speed higher than 5 cm/s; (2) the duration of the sequence ranged from 100 to 200 ms; (3) more than 3 place cells which emitted at least 5 spikes within each sequence. In order to investigate the role of FG- and NFG-cells in theta sequence development, we decoded sequences without either FG- or NFG-cells and cut out the sequences using the same theta cycles as above, defined as 'exFG-sequences' and 'exNFG-sequences' (*Figures 3 and 4*).

To investigate the relationship between sequence development and slow gamma phase precession, we defined FG- and NFG-cell sequences (*Figure 8*). FG-cell sequences are theta sequences with (1) at least one FG-cell and (2) at least firing spikes across three slow gamma cycles for an individual cell. And the 'NFG-cell-dominant sequences' are theta sequences with (1) only NFG-cells and (2) at least firing spikes across three slow gamma cycles for an individual cell.

### Weighted correlation of theta sequences

Weighted correlation was calculated to identify sequential structure within a theta cycle. Decoding probabilities (*prob*) were assigned as the weights of the position estimates to calculate the correlation coefficient between the time (*t*) and the decoding position (*p*) as follows:

$$corr_w\left(t,p;prob\right) = \frac{cov\left(t,p;prob\right)}{\sqrt{cov\left(t,t;prob\right)cov\left(p,p;prob\right)}}$$

and the weighted covariance between time and decoded position $cov\left(t,p;prob\right)$ is as follows:

$$cov\left(t,p;prob\right) = \frac{\sum_i prob_i\left(t_i - m\left(t;prob\right)\right)\left(p_i - m\left(p;prob\right)\right)}{\sum_i prob_i}$$

Weighted means of time and decoded position are as follows:

$$m\left(t;prob\right) = \frac{\sum_i prob_i t_i}{\sum_i prob_i}$$

and

$$m\left(p;prob\right) = \frac{\sum_i prob_i p_i}{\sum_i prob_i}$$

Weighted correlation of each theta sequence was calculated from the decoded probabilities of positions 35 cm behind and ahead of the rat's current location, in a time window of ±30 ms (approximate 1⁄4 theta cycle) before and after the mid-time point of each theta sequence (*Figures 3 and 4*). A significant sweep-ahead structure in the current running direction would produce a large positive correlation, whereas a lack of sequential structure would produce a correlation close to zero.

### Slow gamma phase shift analysis of cell ensemble

Slow gamma phase shift of cell ensemble was estimated as previously described (*Zheng et al., 2016*). The gamma cycle with maximal spiking across all simultaneously recorded cells was defined as cycle 0. Gamma cycles occurring before cycle 0 are represented by decreasing integer values, after by increasing integer values. Incomplete cycles at the beginning or end of the sequence or during immobile (speed <5 cm/s) were excluded from analyses. We estimated the gamma and theta phases of spike times for place cells that exhibited spikes within at least two slow gamma cycles or at least three fast gamma cycles within a single theta cycle. The number of cycles analyzed within each theta sequence was limited to three (cycles –1 to 1) for slow gamma. Two-dimensional histograms of gamma phases and theta phases for spikes from each cycle number were binned into 40 bins and smoothed across nine bins (*Figure 6*).

### Slow gamma phase shift analysis of individual cell

Time-varying slow gamma power (see Detection of gamma episodes) was averaged within each theta cycle. To identify the theta cycle with high slow gamma power, we z-scored time-varying slow gamma power across the theta cycle within each trial. The theta cycle with z-scored time varying power ≥1.5 was defined as slow gamma events. The global 0 phase of slow gamma, set at the trough, determined the phase shift. Slow gamma cycles with at least one spike was defined as active slow gamma cycles (*Figure 7A*). The slow gamma phase difference of a theta cycle with $k$ ($k \geq 2$) adjacent active slow gamma cycles (active cycle) is as follows:

$$\text{slow gamma phase difference} = \frac{\sum_{i=2}^{k}\left(\varphi_i - \varphi_{i-1}\right)}{k - 1}$$

where $\varphi_i$ and $\varphi_{i-1}$ are, respectively, slow gamma phase of a spike in $i$th and ($i$-1)th active cycle. A negative slow gamma phase difference value indicates phase precession, while a positive value indicates phase procession.

In order to test if the negatively biased distribution was robust for slow gamma phase precession, we generated surrogate data and compared their distribution with the real data. A random slow gamma phase within the same cycle was assigned to each spike to keep its firing location constancy, and the distribution of slow gamma phase difference was obtained for this mock data. This procedure was repeated 1000 times to get 1000 distributions and the 95% confident interval of their median (black lines in *Figure 7C–E*). In the analysis, we calculated one-sided, corrected empirical p-values [(r

+ 1)/(n + 1)] by comparing real data to its corresponding shuffle data. Where $r$ is the number values from the shuffled distribution that are either smaller or larger (depending on the hypothesis) than the observed value and $n = 1000$ is the number of shuffling (*Figure 7C–E*).

## Correlation between slow gamma phase precession and weighted correlation

FG- and NFG-cell sequences were included in this analysis (see 'Theta sequences detection' section). For each type of sequences, a scatter plot was made to visualize the relationship between slow gamma phase precession and theta sequence structure, with slow gamma phase shift of each sequence on *x*-axis and its weighted correlation on *y*-axis (*Figure 8A, D, G*). Meanwhile, a two-dimensional histogram was measured to show the distribution of sequences in four quadrants (*Figure 8B, E, H*). We summed up the probability in Quadrant 1 (Q1) and compared it with that of the mock data generated in *Figure 7*. The statistical significance was determined by calculating the one-sided, corrected empirical p-value, with lower p-value indicating significantly higher probability in Q1 of real data than that of the mock data (*Figure 8C, F, I*).

## Histology

At the end of the experiment, histological sections were prepared as follows to verify the tetrodes position. Rats were intraperitoneally injected with a lethal dose of urethane solution. The heart was subsequently perfused with phosphate buffer and formalin to fix the brain tissue. Rat brains were then cut into 30 µm coronal sections and Nissl Staining was performed to determine the location of tetrode in CA1.

## Quantification and statistical analysis

Data analyses were performed using SPSS (IBM), custom Matlab (the Math Work) and R scripts. Paired *t*-test was performed to compare weighted correlation for exFG- and exNFG-sequences during early and late development (*Figure 3H*). Student's *t*-test was performed to compare the mean firing rate, spike counts, place fields size and spatial information for the FG- and NFG-cell (*Figure 5D–G*). One-sample *t*-test was used to test whether the slow gamma phase shift was significantly lower than 0 (*Figure 7B*). Repeated measures ANOVA was performed to test the MVL for FG- and NFG-cell (*Figure 2D*), weighted correlation of theta sequences (*Figure 3E, G*), and running speed (*Figure 3F*) across laps. Generalized linear mixed model was performed to compare weighted correlation among the Raw-, exFG-, and exNFG-sequence. Chi-squared test was performed to compare the proportion of neurons had significant theta phase precession in FG- and NFG-cells (*Figure 2—figure supplement 3*). Kolmogorov–Smirnov test was performed to test the distribution of place field COM (*Figure 5C*) on the track using the Matlab function 'kstest2'. Watson–Williams multi-sample test (circular one-way ANOVA test) was performed to test the for differences in head directions across laps (*Figure 3—figure supplement 1*) using the 'circ_wwtest' function from the CircStat Toolbox. Multi-sample Mardia–Watson–Wheeler test was performed to test the slow gamma phase shift of place cells across successive slow gamma cycles (*Figure 6*) using the R 'circular-package'.

# Acknowledgements

We would like to acknowledge Laura Lee Colgin for her mentorship on some data analysis of slow and fast gamma rhythms. We would also like to acknowledge thank Shuang Meng for help with rat behavioral pre-training; and all Zheng Lab members for helpful discussions.

## Additional information

### Funding

| Funder | Grant reference number | Author |
|---|---|---|
| National Science and Technology Innovation 2030 Major Project of China | 2022ZD0205000 | Chenguang Zheng |
| National Natural Science Foundation of China | T2322021 | Chenguang Zheng |
| National Natural Science Foundation of China | 82271218 | Chenguang Zheng |
| National Natural Science Foundation of China | 12271272 | Lei Wang |
| National Natural Science Foundation of China | 81925020 | Dong Ming |
| National Natural Science Foundation of China | 82371886 | Jiajia Yang |
| National Natural Science Foundation of China | 82202797 | Ling Wang |
| Space Brain Project from Lingang Laboratory | LG-TKN-202204-01 | Jiajia Yang |
| China Postdoctoral Science Foundation | 2022M712365 | Ling Wang |

The funders had no role in study design, data collection, and interpretation, or the decision to submit the work for publication.

### Author contributions

Ning Wang, Software, Formal analysis, Validation, Investigation, Visualization, Methodology, Writing – original draft, Writing – review and editing; Yimeng Wang, Xueling Wang, Nan Zhu, Investigation; Mingkun Guo, Software, Investigation; Ling Wang, Jiajia Yang, Funding acquisition, Investigation; Lei Wang, Conceptualization, Software, Formal analysis, Funding acquisition, Methodology; Chenguang Zheng, Conceptualization, Software, Formal analysis, Supervision, Funding acquisition, Investigation, Methodology, Writing – original draft, Project administration, Writing – review and editing; Dong Ming, Conceptualization, Supervision, Funding acquisition, Investigation, Writing – original draft, Project administration, Writing – review and editing

### Author ORCIDs

Ning Wang ⓘ https://orcid.org/0000-0003-4741-2216
Chenguang Zheng ⓘ https://orcid.org/0000-0002-4782-8732

### Ethics

All experiments were conducted according to the guidelines of the Animal Care and Use Committee of Tianjin University (Approval No. TJUE-2023-142).

Reviewer #2 (Public review): https://doi.org/10.7554/eLife.97334.4.sa1
Author response https://doi.org/10.7554/eLife.97334.4.sa2

## Additional files

### Supplementary files

MDAR checklist

## Data availability

Custom scripts for analysis and visualization have been deposited in GitHub repositories (https://github.com/WNunc/Dynamic_gamma_modulation, copy archived at *WNunc, 2025*).

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
