## [Editor Report · eLife Assessment]

Using electrophysiological recordings in freely moving rats, this **valuable** study investigates the role of gamma oscillations in the development of spatial representations in the hippocampus. Specifically, **solid** evidence supports the claim that distinct gamma oscillatory inputs contribute to the emergence of 'theta sequences', which encode the animal's ongoing trajectory. This study will be of interest to neuroscientists working in the fields of spatial navigation and neuronal dynamics.

---

## [Referee Report · Reviewer #2 (Public review)]

This manuscript addresses an important question which has not yet been solved in the field, what is the contribution of different gamma oscillatory inputs to the development of "theta sequences" in the hippocampal CA1 region. Theta sequences have received much attention due to their proposed roles in encoding short-term behavioral predictions, mediating synaptic plasticity, and guiding flexible decision-making. Gamma oscillations in CA1 offer a readout of different inputs to this region and have been proposed to synchronize neuronal assemblies and modulate spike timing and temporal coding. However, the interactions between these two important phenomena have not been sufficiently investigated. The authors conducted place cell and local field potential (LFP) recordings in the CA1 region of rats running on a circular track. They then analyzed the phase locking of place cell spikes to slow and fast gamma rhythms, the evolution of theta sequences during behavior and the interaction between these two phenomena. They found that place cell with the strongest modulation by fast gamma oscillations were the most important contributors to the early development of theta sequences and that they also displayed a faster form of phase precession within slow gamma cycles nested with theta.

Comments on revisions:

Several important shortcomings were noted in the original manuscript. These have been addressed in this revised version with the addition of multiple new analysis, controls and clarifications. The revised manuscript has been significantly improved and its conclusions are adequately supported by the results presented.

---

## [Author Response]

The following is the authors’ response to the previous reviews

**Reviewer #1 (Public review):**

This study presents evidence that a special group of place cells, those tuned to fast-gamma oscillations, play a key role in theta sequence development. How theta sequences are formed and developed during experience is an important question, because these sequences have been implicated in several cognitive functions of place cells, including memory-guided spatial navigation. The revised version of this paper has been significantly improved. Major concerns in the previous round of review on technical and conceptual aspects of the relationship between gamma oscillations and theta sequences are addressed. The main conclusion is supported by the data presented.

**Reviewer #2 (Public review):**
The authors have conducted new analysis to address the issues I and the other reviewers raised in our original revision. As a result, the revised manuscript has been substantially improved.

We thank the two reviewers for their positive comments.

**Recommendations for the authors:**

**Reviewer #2 (Recommendations for the authors):**
There are, however, still a few remaining issues that need further clarification.- Despite the authors explanation and comparison with Kitanishi et al., 2015, Neuron, I still find that the reduced number of significantly gamma phase-locked cells is at odds with most previous reports (e.g., Csicvari et al., 2003; Colgin et al., 2009; Belluscio et al., 2012; Schomburg et al., 2014; Cabral et al., 2014; Fernandez-Ruiz et al., 2017; Lopes dos Santos et al., 2018). There can be several issues to explain this difference, like the choice of LFP reference channel. The authors should at least acknowledge this difference in the text.

We thank the reviewer for this suggestion. We discussed the potential reasons causing the different proportion of gamma phase locked cells in the Discussion (lines367-380).

- The new Figure R2 is very useful and should be included in the manuscript. It would be even better to expand the frequency range to higher frequencies to show where the maximum peak is. Still, the potential contribution of spike leakage should be acknowledged. While I agree that it will not account for all fast gamma spike modulation, it is certainly a contributing factor. A further evidence of this is that the coupling strength seems to keep increasing towards supra gamma frequency range in Fig R2. This is to be expected given that the authors have used the local LFP from the same tetrode where cells were recorded, which is never a good practice.

We thank the reviewer for this suggestion. Now the Fig R2 has been moved to the manuscript as a part of Figure 2-figure supplement 2 (lines133-135). In terms of the contribution of spike leakage by using the local LFP, we also detected FG-cells by using LFP from a different tetrode, i.e. the central one of the bundle that located in the cell body layer, and found approximate proportion of FG-cells which phase locked to ~75Hz (Fig R3, now the Figure 2-figure supplement 2C-F). Thus, we think using the local LFP would not affect the main conclusion and we decide to keep the original results. We also acknowledged the potential contribution of spike leakage in the Discussion (lines 372-377).

- From the authors answer I understand that recordings were almost exclusively conducted from the deep CA1 pyramidal layer. This would preclude any meaningful interpretation of the deep/ superficial differences in the distribution of FG and NFG cells. This is not a crucial point for the paper but needs to be acknowledged.

We thank the reviewer for this suggestion. We acknowledged the meaningful interpretation of the deep/ superficial differences in the distribution of FG- and NFG-cells in the Discussion (lines 380-386).

- I am afraid that the authors interpreted my comment about authorship in the opposite way that I intended. I meant that the usual practice is that the last author of the manuscript is the person who has been the main intellectual driver of the work, not the most senior one necessarily. I guess that is Dr. Zheng not Dr. Ming. However, I leave this decision to the discretion of the authors.

We thank the reviewer for this rigorous consideration. Dr. Ming and Dr. Zheng were both the main intellectual drivers of this work. Therefore, we decide to keep the current authors in the manuscript.